# Clinical outcome of FIGO 2018 stage IB3/IIA2 cervical cancer treated by neoadjuvant chemotherapy followed by radical surgery due to lack of radiotherapy equipment: A retrospective comparison with concurrent chemoradiotherapy

Jing Zeng[1☯], Peisong Sun[1☯], Quanhong Ping[1], Shan Jiang[2], Yuanjing Hu🆔[1] *

1 Tianjin Clinical Research Center for Obstetrics and Gynecology, Department of Gynecologic Oncology, Tianjin Central Hospital of Gynecology and Obstetrics, Affiliated Hospital of Nankai University, Tianjin, China, 2 School of Mechanical Engineering, Tianjin University, Tianjin, China

☯ These authors contributed equally to this work.
* huyuanjingzxfck@163.com

## Abstract

This study aimed to assess neoadjuvant chemotherapy's clinical outcomes such as efficacy, toxicity, and survival outcomes followed by radical hysterectomy ((NACT-RS) among women with cervical cancer stage IB3 and IIA2, by comparing concurrent chemoradiotherapy (CCRT) and NACT-RS. The study retrospectively reviewed patients with (2018 FIGO) stage IB3 and IIA2 cervical cancer who received preoperative neoadjuvant chemotherapy followed by NACT-RS or concurrent chemoradiotherapy (CCRT). The outcome measures were the 5-year survival and complication rates between the two groups. The median follow-up was 75 months. In total, 218 patients had stage IIA2, 136 patients had stage IB3, 201 patients received CCRT, and 153 patients received preoperative NACT-RS. In the CCRT group, the incidence of early complications (myelosuppression, gastrointestinal and urinary) was higher compared with that in the NACT-RS group (76.1 vs. 26.1%; $p < 0.001$, respectively). There was no significant difference between the two study groups concerning late complications. Five-year PFS was 79.9% and 85.5% in the NACT-RS and CCRT groups, respectively ($p = 0.093$). Five-year OS was 86.9% and 85.5% in the NACT-RS and CCRT groups, respectively ($p = 0.97$). In the multivariate clinicopathologic characteristics analysis for OS, initial tumor size > 4.3 cm (HR 5.11; $p < 0.001$), AC/ASC (HR 1.89; $p = 0.02$), histologic grade 2–3 (HR 2.25; $p = 0.04$), and 2018 FIGO stage IIA2 (HR 8.67; $p < 0.001$) were independent risk factors. The survival of patients with stage IB3 and IIA2 cervical cancer treated with NACT-RS was similar to that of patients treated with CCRT without increasing side effects.

**Data Availability Statement:** All relevant data are within the paper and its Supporting Information files.

**Funding:** This study was supported by the Health Science and Technology Project of Tianjin (No. KJ20149 and ZC20111). Jing Zeng designed the study and collected and analyzed the data. Quanhong Ping prepared the manuscript, and Peisong Sun decided to publish it.

**Competing interests:** The authors have declared that no competing interests exist.

## Introduction

Cervical cancer is the fourth most frequent cancer and the second most common in low- and middle-income countries [1]. Approximately 80% of new cases and 85% of deaths occur in developing countries, and more than 70% of them are diagnosed at the locally advanced stage [2]. Local advanced cervical cancer (LACC) generally refers to the 2009 International Federation of Gynecology and Obstetrics (FIGO) stage IB2-IVA cervical cancer, while the narrow sense of LACC refers typically to (2009 FIGO) stage IB2 and IIA2 cervical cancer. As a standard treatment for advanced cervical cancer, the 5-year overall survival rate of concurrent chemoradiotherapy (CCRT) is up to 65% [3]. However, the standard treatment for LACC is still controversial, and its survival rate is still frustrating [4]. In 1982, Frei [5] first proposed the concept of neoadjuvant chemotherapy (NACT). In the 1990s, to improve the resection rate of LACC patients and reduce the incidence of postoperative recurrence and metastasis, he also proposed the concept of neoadjuvant chemotherapy for locally advanced cervical cancer. However, the clinical effect of NACT in patients with LACC before the operation is also controversial.

In 2018, The International Federation of Gynecology and Obstetrics (FIGO) revised staging criteria and guidelines, recommending concurrent chemoradiation treatment for stages IB3 and IIA2 (2009 FIGO stage IB2 and IIA2) cervical cancers. In addition, neoadjuvant chemotherapy followed by radical surgery (NACT-RS) is a treatment option for stage IB3 and IIA2 when radiotherapy equipment is unavailable [6, 7].

The purpose of this study was to assess the efficacy, toxicity, and survival outcomes of NACT-RS among patients with cervical cancer stage IB3 and IIA2. Therefore, we conducted a retrospective study to compare NACT-RS vs. CCRT in patients with stage IB3 and IIA2 cervical cancer.

## Patients and methods

Our study retrospectively reviewed clinical results of patients with cervical cancer staged IB3 and IIA2 treated in Tianjin Central Hospital of Gynecology Obstetrics (Tianjin, China) between January 2011 and December 2016 according to the 2018 FIGO classification. Cervical biopsies were used for histological confirmation. In our research, the preoperative cervical biopsy included squamous cell carcinoma (SCC), adenocarcinoma, or adenosquamous carcinoma (AC/ASC). Cases with an interval between the end of treatment and the beginning of the study lower than 6 months were excluded from the study.

In our center, the preferred option recommended for the initial treatment of cervical cancers with operable tumors (≥4 cm in diameter) at stages IB and IIA is concurrent chemoradiation. However, NACT followed by radical hysterectomy and lymphadenectomy was also a treatment option due to unavailability in our center (due to replacement) of radiotherapy equipment between January 2014 and December 2016.

In the NACT group, two cycles of paclitaxel and cisplatin (TP)/paclitaxel and carboplatin (TC) (135–155 mg/m$^2$, day 1, and 60 mg/m$^2$, respectively) (area under the concentration-time curve 5.0–7.5, day 1) administered at a 21-day interval [8]. The patients underwent radical hysterectomy and lymphadenectomy by laparotomy or laparoscopy within an average period of 7 days following NACT. Frozen sections analysis was performed for the external iliac lymph nodes in the lymphadenectomy. The cases of positive lymph nodes in frozen sections analysis or postoperative pathological analysis were excluded from our study. Patients in the CCRT group were treated with external pelvic radiotherapy and brachytherapy with concomitant chemotherapy. External beam radiotherapy (EBRT) was delivered to the whole pelvis with a total dose of 45 Gy in 25 fractions using the four-field box technique with $^{60}$Co external-beam.

High-dose-rate brachytherapy with a total dose of 35–45 Gy (2-Gy daily fractions (EQD2), assuming an α/β ratio of 10 Gy) at point A was performed weekly for 4 consecutive weeks using $^{192}$Ir sources. The total equity effective prescription for a total point A dose was 80–90 Gy EQD2. Concurrent chemotherapy was started at the beginning of EBRT. Patients received three or four cycles of cisplatin at 50 mg/m$^2$ at 21-day intervals and 1 day of paclitaxel (135–175 mg/m$^2$).

Periodic follow-up visits were scheduled. Specific examinations were performed, including abdominopelvic magnetic resonance imaging (MRI) or computed tomography (CT).

NACT response was evaluated by pelvic examination and ultrasound before surgery and was recorded as complete response (tumor completely disappeared), partial response (tumor size reduced more than 50%), stable disease (tumor size reduced less than 50%, or increased less than 25% without new lesions), or progressive disease (tumor size increased more than 25% or had new lesions), according to the World Health Organization (WHO) solid tumor efficacy criteria [9]. All patients who received NACT received surgery. We assessed safety in terms of complications, according to the Chassagne glossary [10]. Early complications were defined as complications occurring during or within 2 months of treatment completion. Late complications were defined as any complications appearing not before 91 days after the end of treatment and were prospectively scored using Common Terminology Criteria of Adverse Event (CTCAE v5.0).

In the CCRT group, after the treatment was completed, patients' follow-up included a pelvic exam performed to evaluate the disease status according to the National Comprehensive Cancer Network (NCCN) criteria. Late toxicities arising three months after the end of treatment were evaluated according to the Radiation Therapy Oncology Group/European Organization of Research and Treatment of Cancer (RTOG/EORTC) late toxicity criteria.

Disease-free survival (DFS) was defined as the duration between the beginning of treatment and the date of the first documented evidence of relapse at any site (local recurrence, metastasis, or both) or death. The overall survival duration (OS) was calculated from the beginning of treatment to death. The Kaplan–Meier technique was used to calculate the proportion of OS and DFS, and the log-rank test was used to assess the statistical significance. Pathology results within the NACT-RS group, complications, and recurrence patterns were analyzed using the chi-square test to compare patient and tumor parameters. Log-rank test, hazard ratio (HR), and Cox model hazard regression analysis with 95% confidence intervals (CIs) were applied for univariate and multivariate analyses of OS for all patients. Statistical significance was defined as a $p$-value <0.05. The SPSS version 19.0 software package (IBM, Chicago, IL, USA) was used for statistical analyses.

The study was conducted following the Declaration of Helsinki (revised in 2013). The study was approved by the institutional Ethics Review Board of Tianjin Central Hospital of Gynecology and Obstetrics (No. 2020KY066), and written informed consent was obtained from all the patients. The data were obtained from medical records in a fully anonymized and de-identified manner, and all authors had access to identifying information.

## Results

We identified 354 patients who met the inclusion criteria; NACT-RS treated 153 and CCRT 201 patients. Patient and tumor characteristics are summarized in Table 1. The median follow-up was 75 months [6–104]. The mean age of these patients was 46 years (range 24–68 years). The mean initial tumor size was 4.3 cm (range 4–8 cm). Patients in the CCRT group were at a clinical disadvantage as compared with those in the NACT-RS group, with a higher incidence of AC/ASC (18.9% vs. 4.6%; $p < 0.001$) and stage IIA2 cancer (77.6% vs. 40.5%; $p < 0.001$).

**Table 1. Patient and tumor characteristics.**

| Characteristic | NACT-RS (n = 153) | CCRT (n = 201) | p-value |
|---|---|---|---|
| Age mean (range) | 46.6±8.9 (25–68) | 47.11±8.7 (24–66) | 0.591 |
| Anemia before treatment n (%) | 17 (11.1) | 32 (15.9) | 0.126 |
| Smoking > 10 cigarettes/day | 10 (6.5) | 11 (5.5) | 0.223 |
| Initial tumor size in cm mean (range) | 4.29 ± 0.62 (4–6) | 4.33 ± 0.7 (4–8) | 0.213 |
| Histology n (%) | | | <0.001 |
| SCC | 146 (95.4) | 163 (81.1) | |
| AC/ASC | 7 (4.6) | 38 (18.9) | |
| Histologic grade n (%) | | | 0.064 |
| G1 | 38 (24.8) | 66 (32.8) | |
| G2-3 | 115 (75.2) | 135 (67.2) | |
| FIGO 2018 stage n (%) | | | <0.001 |
| Ib3 | 91 (59.5) | 45 (22.4) | |
| IIa2 | 62 (40.5) | 156 (77.6) | |

NACT-RS: Neoadjuvant chemotherapy followed by radical surgery; SCC: Squamous cell carcinoma, AC/ASC: Adenocarcinoma or adenosquamous carcinoma, FIGO: International Federation of Gynecology and Obstetrics.

The patients showed different degrees of response after chemotherapy in the NACT-RS group. Clinical response (complete or partial response) was determined in 138/153 (90.2%) patients. Sixty-four patients (46.4%) experienced complete remission, with 74 patients (53.6%) experiencing partial responses. According to the Querleu–Morrow classification [11] of radical hysterectomy, 36 patients underwent laparoscopy interventions: two patients with type B2 and 34 patients with type C2. In total, 117 patients underwent laparotomy interventions: seven patients with type B2 and 110 patients with type C2. After the surgical procedure, the tumor size median diameter was 2.3 cm (range 1–7.5 cm). Among patients (153) in the NACT-RS group, seven (4.5%) had parametrial invasion, 74 (48.3%) lymph-vascular invasion, three (2%) positive surgical margin, and 46 (30%) deep cervical stromal invasion >50%. Depending on the presence of high-risk factors (parametrial invasion and positive surgical margins) and potentially important risk factors (Sedilis Criteria), only 15 of 153 (9.8%) patients received additional external beam radiation within one month after surgery in the NACT-RS group, in the form of conformal radiation therapy. The delivered dose to the pelvis was 45 Gy. Meanwhile, 10/153 (6.5%) patients received postoperative chemoradiotherapy. TC was used for AC/ASC, and TP was used for squamous cell carcinoma (135–155 mg/m$^2$, day 1, and 60 mg/m$^2$, respectively; area under the concentration-time curve 5.0–7.5, day 1). The median number of cycles of chemotherapy was 4 [1–6].

Response (complete or partial) and non-response (stable disease or progressive disease) groups were assigned according to the pathology results of surgical specimens, including parametrial invasion, lymph vascular invasion, positive surgical margin, and deep cervical stromal invasion (more than 50%). It was shown that parametrial invasion, lymph vascular invasion, and deep stromal invasion (more than 50%) were significantly reduced in the response group (3% vs. 20%, $p = 0.021$; 43.5% vs. 93.3%, $p < 0.001$; 26.1% vs. 66.7%, $p = 0.002$, respectively) for patients with NACT-RS (Table 2).

Early complications during or within two months of treatment completion are listed in Table 3. As for grade 1–2 complications, 17 patients with NACT-RS treatment had varying degrees of myelosuppression, including anemia, thrombocytopenia, and neutropenia, compared to 76 patients with CCRT treatment (9.2% vs. 37.8%, respectively; $p < 0.001$). Eleven

**Table 2. Comparison of pathology results within NACT-RS group.**

| Result (n) | Response (n = 138) | No- response (n = 15) | *p*-value |
|---|---|---|---|
| Parametrial invasion | 4 | 3 | 0.021 |
| Lymph vascular invasion | 60 | 14 | <0.001 |
| Positive surgical margin | 2 | 1 | 0.268 |
| Deep cervical stromal invasion >0.5 | 36 | 10 | 0.002 |

(11) patients that received NACT-RS treatment had varying degrees of gastrointestinal reactions, including vomiting, diarrhea, transient GI bleeding, and intestinal obstruction, compared to 93 patients that received CCRT treatment (11.1% vs. 46.3%, respectively; $p < 0.001$). In addition, 16 patients had varying degrees of urinary incontinence, urgency, and dysuria, compared to 78 CCRT-treated patients (10.5% vs. 38.8%; $p < 0.001$). Moreover, three patients had grade 3 complications in the NACT-RS group and 37 in the CCRT group (3% vs. 18.4%, respectively; $p < 0.001$).

The main toxicities associated with CCRT treatment were grade 1–2, including 35 (17.4%) pelvic lymphedema, 30 (14.9%) symptomatic vaginal stenosis, 18 (9%) persistent GI bleeding, and four (2%) hematuria (Table 3). For the grade 3 complications, resection of part of the intestine was performed in three (1.5%) cases due to enteric necrosis. As for NACT-RS treatment, there were 58 (37.9%) grade 1–2 complications, including 40 (26.1%) cases of urinary problems (urgency, dysuria, and ureterohydronephrosis), four (2.6%) cases of symptomatic vaginal stenosis, and 14 (9.2%) cases of pelvic lymphedema. Moreover, there were three (2%) grade 3 complications, including one (0.7%) case of definitive colostomy and two (1.3%) cases of pelvic lymphedema. However, there was no significant difference between the two study

**Table 3. Early and late complications according to the Chassagne glossary.**

| Result (n) | NACT-RS (n = 153) | CCRT (n = 201) | *p*-value |
|---|---|---|---|
| **Early complications, n (%)** | 40 (26.1) | 153 (76.1) | < 0.001 |
| **Grade 1–2** | 38 (24.8) | 123 (61.2) | < 0.001 |
| myelosuppression | 17 (9.2) | 76 (37.8) | < 0.001 |
| Gastrointestinal | 11 (11.1) | 93 (46.3) | < 0.001 |
| Urinary | 16 (10.5) | 78 (38.8) | < 0.001 |
| **Grade 3** | 3 (2) | 37 (18.4) | < 0.001 |
| myelosuppression | 0 | 12 (6) | 0.001 |
| Gastrointestinal | 3 (2) | 19 (9.5) | 0.003 |
| Urinary | 0 | 6 (3) | 0.032 |
| **Late complications, n (%)** | 61 (39.9) | 89 (44.3) | 0.235 |
| **Grade 1–2** | 58 (37.9) | 87 (43.3) | 0.182 |
| Gastrointestinal | 0 | 18 (9) | < 0.001 |
| Urinary | 40 (26.1) | 4 (2) | < 0.001 |
| Symptomatic vaginal stenosis | 4 (2.6) | 30 (14.9) | < 0.001 |
| pelvic lymphedema | 14 (9.2) | 35 (17.4) | 0.018 |
| **Grade 3** | 3 (2) | 3 (1.5) | 0.523 |
| Gastrointestinal | 1 (0.7) | 3 (1.5) | 0.42 |
| pelvic lymphedema | 2 (1.3) | 0 | 0.186 |

CCRT, concomitant chemotherapy, and radiotherapy; NACT-RS, neoadjuvant chemotherapy followed by radical surgery.

NOTE. Some patients had more than one complication.

**Table 4. Pattern of recurrence.**

| Recurrence Site | NACT-RS (n = 153) | CCRT (n = 201) | p-value |
|---|---|---|---|
| **Local, n** | 8 (5.2%) | 4 (2%) | 0.086 |
| lower vaginal | 4 (2.6%) | 1 (0.5%) | |
| parametrial | 3 (2%) | 3 (1.5%) | |
| bladder | 1 (0.7%) | 0 | |
| **Distant, n** | 18 (11.8%) | 24 (11.9%) | 0.548 |
| Supraclavicular lymph node metastasis | 5 (3.3%) | 3 (1.5%) | |
| Isolated pulmonary metastatic (≤3 lesions), n | 3 (2%) | 6 (3%) | |
| Multiple pulmonary metastatic (>3 lesions), n | 10 (6.5%) | 15 (7.5%) | |
| **Local plus distant** | 1 (0.7%) | 2 (1%) | 0.603 |

groups concerning late complications. In addition, no grade 4 complication was observed in either treatment group.

The median follow-up duration was 75 months (4–104). No patients were lost to follow-up. In total, 12 patients (3.4%) had only local recurrence. Forty-two patients (11.9%) experienced only distant relapse. Three patients (0.8%) experienced relapse (local and distant). The pattern of recurrence in the two groups is shown in Table 4. There was no significant difference between the two study groups in the recurrence pattern. Moreover, the recurrence rate (17.4%) of NACT clinical responders compared with (33.3%) of non-responders was lower but without statistical significance ($p = 0.127$). The 5-year PFS was 79.9% and 85.5% in the NACT-RS and CCRT groups, respectively ($p = 0.093$; Fig 1). Moreover, the 5-year OS was 86.9% and 85.5% in the NACT-RS and CCRT groups, respectively ($p = 0.973$; Fig 2). In the multivariate clinicopathologic characteristics analysis for OS, initial tumor size >4.3 cm (HR 5.11; $p < 0.001$), AC/ASC (HR 1.89; $p = 0.02$), histologic grade 2–3 (HR 2.25; $p = 0.04$), and 2018 FIGO stage IIA2 (HR 8.67; $p < 0.001$) were independent risk factors (Table 5).

## Discussion

NACT before surgery or radiotherapy has been researched as a potential therapeutic method for larger (FIGO stages IB-IIA, tumor size ≥4 cm) or LACC. The underlying advantages of NACT are its effectiveness in shrinking tumors, controlling micrometastases, and increasing the rate of obtaining wider, uninvolved resection margins, and thereby avoiding adjuvant radiotherapy [12]. So far, many clinical trials have been conducted on chemotherapy combined with radiotherapy and surgery. Based on the results of five large randomized clinical trials, the National Cancer Institute (NCI) suggested in 1999 that platinum-based concurrent chemoradiotherapy (CCRT) should be the standard treatment for LACC [13]. The recent result of a large-scale clinical study for stage IB2-IIB cervical carcinoma demonstrated no difference in 5-year OS between NACT-RS and CCRT. Compared to CCRT, patients with stage IB2 trended for better results in NACT-RS [14]. A single-center, Phase III randomized controlled trial reported that CCRT resulted in superior DFS than ACR-RS in locally advanced cervical cancer. However, there was also no difference in 5-year OS between the two groups (75.4% vs. 74.7%; $p = 0.87$) [15]. In this study, the 5-year PFS and OS in patients with 2018 FIGO stage IB3 or IIA2 showed no significant difference in the NACT-RS and CCRT groups (79.9% vs. 85.5%, respectively; $p = 0.093$; 86.9% vs. 85.5%, respectively; $p = 0.97$). So far, there is no consensus on the standard treatment of LACC globally, and there is a great controversy on the application of NACT in LACC. Therefore, CCRT is the first choice for stage IB3 and IIA2 cervical cancer. NACT-RS is an alternative treatment for hospitals without CCRT in the 2018 FIGO staging criteria and management [7].

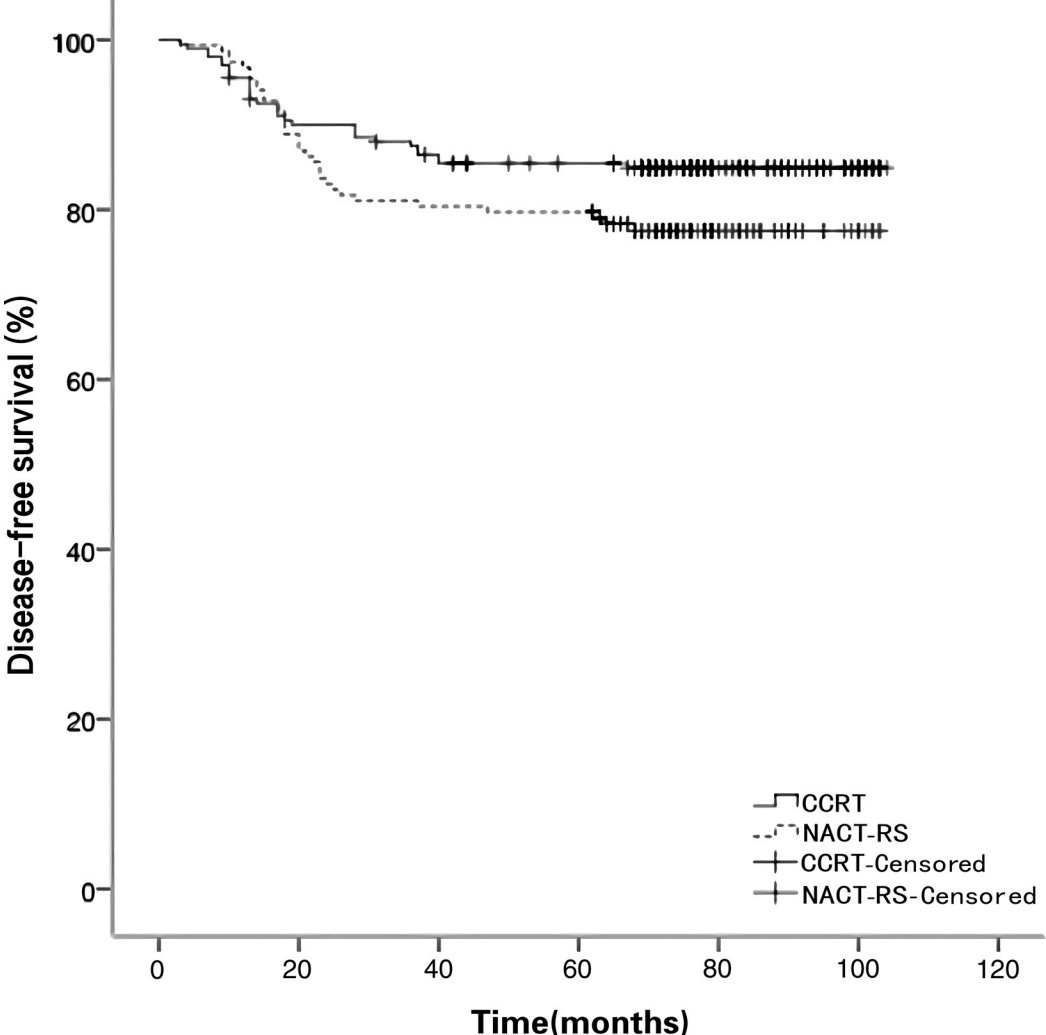

**Fig 1. Disease-free survival curves for patients in the NACT-RS and CCRT groups.** CCRT, concomitant chemotherapy, and radiotherapy; NACT-RS, neoadjuvant chemotherapy followed by radical surgery.

Nonetheless, some studies failed to identify a survival benefit associated with the use of NACT compared to standard treatments [16–18]. It was suggested that only responders to NACT followed by surgery would benefit [19]. As Hu et al. reported, NACT significantly improves 5-year DFS, while mortality and recurrence rates are decreased. Furthermore, patients with 2009 FIGO IB2 and NACT responders greatly benefit from NACT [20]. The recurrence rate (5.2%) of NACT clinical responders was significantly decreased, with a significant improvement in IB tumors (2–5 cm in size) [20]. However, for those who do not respond to NACT, the delay in curative treatment, the development of radio-resistant cellular clones, and cross-resistance with radiotherapy should be considered important disadvantages of NACT [21]. In our research, 15 of 153 (9.8%) patients had no clinical response to NACT. The recurrence rate (17.4%) of NACT clinical responders compared with (33.3%) of non-responders was decreased, but without significant difference ($p = 0.127$). Previous studies concluded that using NACT reduced the need for adjuvant radiotherapy [22, 23]. Neoadjuvant chemotherapy before surgery was shown to reduce the need for adjuvant radiotherapy (OR 0.57; 95%

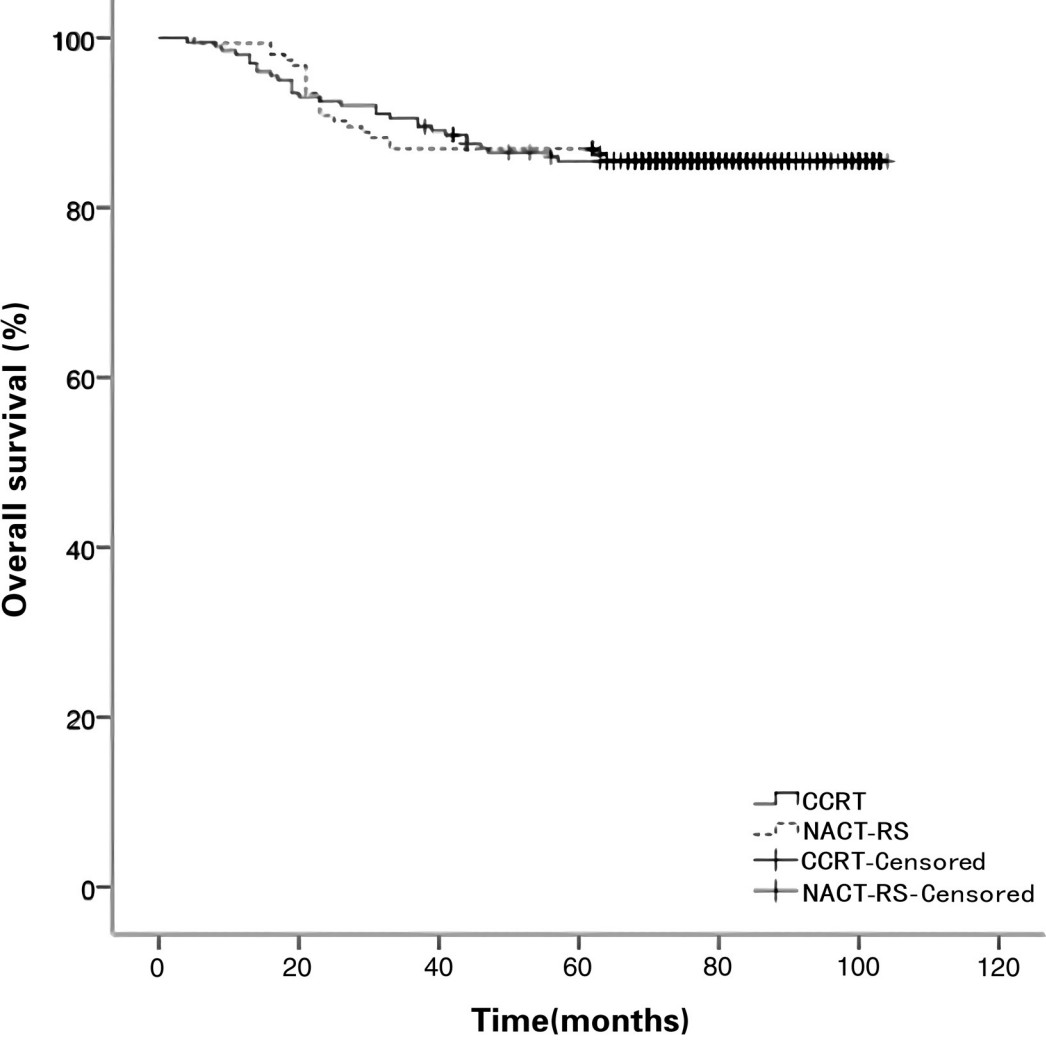

**Fig 2. Overall survival curves for patients in the NACT-RS and CCRT groups.** CCRT, concomitant chemotherapy, and radiotherapy; NACT-RS, neoadjuvant chemotherapy followed by radical surgery.

CI 0.33–0.98). It has been suggested that NACT before surgery might help avoid complications of adjuvant radiotherapy [24, 25]. NACT may reduce intermediate-risk factors, including large tumor size (the longest diameter on a surgical specimen greater than 4 cm) and lymphovascular space invasion (greater than one-third stromal invasion) [20, 26, 27]. In the present study, parametrial invasion, lymph vascular invasion, and deep stromal invasion (greater than 0.5) were significantly reduced in response group (3% vs. 20%; $p = 0.021$; 43.5% vs. 93.3%; $p < 0.001$; 26.1% vs. 66.7%; $p = 0.002$, respectively) for NACT-RS-treated patients. The NACT was effective and enabled avoiding adjuvant radiotherapy for NACT clinical responders. There was no significant difference between the NACT-RS and CCRT groups in the recurrence (distant and/or local).

Although the early complications involving the myelosuppression, gastrointestinal, and urinary were higher in the CCRT group, there was no significant difference in the late complications, indicating their resolution in most patients. Regarding long-term toxicity, the main complications in the NACT-RS group were grade 1–2 and were mostly represented by urinary

**Table 5. Univariate and multivariate analysis of overall survival of all patients.**

| Characteristic | Univariate analysis | Multivariate analysis | |
|---|---|---|---|
| | *p*-value | *p*-value | Hazard ratio (95% CI) |
| Age >46 years | 0.65 | 0.72 | 1.112 (0.62–1.99) |
| Anemia before treatment n (%) | 0.57 | 0.47 | 0.728 (0.31–1.73) |
| Smoking more than10 cigarettes/day | 0.69 | 0.98 | 1.01 (0.55–1.86) |
| Initial tumor size >4.3 cm | 0.05 | < 0.001 | 5.11 (2.16–12.09) |
| AC/ASC | 0.04 | 0.02 | 1.89 (1.1–3.25) |
| Histologic grade G2-3 | 0.02 | 0.04 | 2.25 (1.04–4.85) |
| FIGO 2018 stage (IB3 vs. IIA2) | 0.001 | < 0.001 | 8.67 (3.37–22.28) |
| Treatment (NACT-RS vs. CCRT) | 0.1 | 0.12 | 1.63 (0.88–3.03) |

AC/ASC, Adenocarcinoma or adenosquamous carcinoma; FIGO, International Federation of Gynecology and Obstetrics; NACT-RS, neoadjuvant chemotherapy followed by radical surgery; CCRT, concomitant chemotherapy and radiotherapy.

complications. Because most patients received surgery of type C2 [11] (no preservation of autonomic nerves), that may increase the risk of urinary dysfunction, such as urinary incontinence, urgency, dysuria, and abnormal sensation. Nonetheless, the treatment tolerance of the two groups and the complications were acceptable.

Among the cervical cancer risk factors examined in multivariate analysis, the 2018 FIGO stage IIA2 compared with stage IB3 was one of the substantial factors. Another decisive factor was the initial tumor size greater than 4.3 cm. Bulky tumor volumes characterized LACC. Several studies reported that the 5-year OS of IB2-IIA patients with tumors diameter greater than 4 cm is 30%–60% with the surgical intervention [25, 28, 29]. Our evaluation of NACT-RS was consistent with tumor volume reduction. However, NACT-RS did not decrease the risk factors or offer survival benefits compared with CCRT. Therefore, patients with bulky cervical tumor volumes should be assessed before treatment to benefit from the choices, including CCRT and NACT-RS. The former approach increased the local control rate of the disease. In addition, the latter approach provided the advantages of surgery, and the NACT decreased the risk of metastasis without adjuvant radiotherapy [30, 31].

Histologic grade is a surgical-pathology parameter for predicting patient outcomes [32]. The degree of infiltration is considered necessary for determining the adjuvant treatment after radical hysterectomy [33, 34]. The current study also found that patients with grade 2–3 (intermediate/low differentiation) pathology had poor DFS and OS. According to the surgical-pathology staging and scoring system for cervical cancer [35], a histologic grade risk factor for prognosis is used for defining surgical-pathology stages.

Furthermore, 45 patients with AC/ASC had low 5-year OS. Previous studies have also suggested that the AC/ASC patients with the same FIGO stage have poorer prognostic outcomes than SCC [36, 37]. According to the present guidelines [7, 38], the therapeutic strategy recommended for SCC and AC/ASC is the same. Surgery and radiotherapy are recommended as the therapeutic schedule for early-stage cervical cancer. In our study, 38 patients were treated by CCRT, and NACT-RS only treated seven. However, a population-based analysis of about 2,773 patients with early-stage adenocarcinoma (IB-IIA) found that surgery remains the optimal local treatment modality for these patients [39]. Another retrospective study of about 3,102 patients with cervical adenocarcinoma using the Epidemiology and End Results (SEER) database also found that surgery is an independent favorable prognostic factor for early stages

patients [40]. Therefore, it is significant to make new clinical treatment strategies for cervical AC/ASC.

This was a small-sample retrospective study with a particular inherent bias. The clinical-pathology dissimilarities between the types of cervical cancer (SCC vs. AC/ASC) presented differences that could interfere with the statistics of the obtained results. The number of IIA2 (more advanced stage) in the CCRT group was higher than in the NACT-RS group (156 in CCRT vs. 62 in NACT-RS) that may introduce a bias to the same results between the two groups. Due to the limited number of cases in a single institutional study, the analysis could not include patients (stages IB3 and IIA2) with RS alone. Additionally, our research used the four-field box technique with $^{60}$Co external-beam and two-dimensional brachytherapy. New technology such as intensity-modulated radiotherapy (IMRT) and image-guided brachytherapy (IGBT) could significantly reduce the toxicity of radiotherapy [41]. In all, a larger prospective randomized trial with a multicenter is required to confirm these results.

## Conclusion

The survival of patients with stage IB3 and IIA2 cervical cancer treated with NACT-RS was similar to that of patients treated with CCRT without increasing side effects. Therefore, in the absence of radiotherapy equipment, preoperative NACT followed by radical surgery is viable for patients with stage IB3 and IIA2 cervical cancer.

## Supporting information

**S1 File. Study's minimal underlying data set.**
(SAV)

## Author Contributions

**Conceptualization:** Peisong Sun.

**Data curation:** Peisong Sun.

**Formal analysis:** Peisong Sun.

**Funding acquisition:** Peisong Sun.

**Investigation:** Jing Zeng.

**Methodology:** Jing Zeng.

**Project administration:** Yuanjing Hu.

**Resources:** Quanhong Ping.

**Software:** Quanhong Ping.

**Supervision:** Yuanjing Hu.

**Validation:** Shan Jiang.

**Visualization:** Shan Jiang.

**Writing – original draft:** Jing Zeng.

**Writing – review & editing:** Jing Zeng.

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
