## [Decision Letter · Decision Letter 0]

4 Nov 2021

PONE-D-21-13754

Clinical outcome of FIGO 2018 stage IB3/IIA2 cervical cancer submitted to neoadjuvant chemotherapy followed by radical surgery in lack of radiotherapy equipment: A retrospective comparison with concurrent chemoradiotherapy

PLOS ONE

Dear Dr. Yuanjing Hu,

Thank you for submitting your manuscript to PLOS ONE. After careful consideration, we feel that it has merit but does not fully meet PLOS ONE’s publication criteria as it currently stands. Therefore, we invite you to submit a revised version of the manuscript that addresses the points raised during the review process.

We look forward to receiving your revised manuscript.

Kind regards,

Ramon Andrade De Mello, MD, PhD, FACP

Academic Editor

PLOS ONE

Journal Requirements:

"This study was supported by the Science and Technology Project of Tianjin (No.18YFZCSY01300) and Health Science and Technology Project of Tianjin (No. ZC20111 and KJ20149)."

7. Your ethics statement should only appear in the Methods section of your manuscript. If your ethics statement is written in any section besides the Methods, please delete it from any other section. 

8. Please upload a copy of Figures 1 and 2, to which you refer in your text on page 13. If the figure is no longer to be included as part of the submission please remove all reference to it within the text.

Reviewers' comments:

Reviewer's Responses to Questions

**Comments to the Author**

1. Is the manuscript technically sound, and do the data support the conclusions?

Reviewer #1: Yes

Reviewer #2: Yes

2. Has the statistical analysis been performed appropriately and rigorously? 

Reviewer #1: Yes

Reviewer #2: Yes

3. Have the authors made all data underlying the findings in their manuscript fully available?

Reviewer #1: Yes

Reviewer #2: Yes

4. Is the manuscript presented in an intelligible fashion and written in standard English?

Reviewer #1: Yes

Reviewer #2: Yes

5. Review Comments to the Author

Reviewer #1: For the critical analysis of the article, I suggest the evaluation of the following topics:

- Address the clinical-pathological differences between the types of cervical cancer (SCC x AC / ASC) in the discussion, as the groups present statistical differences in this regard, which may interfere with the results obtained.

-Evaluate include in the discussion the recent data from the following article: DOI: 10.1038 / s41598-021-86786-y

Reviewer #2: Well writen, considering its a common cancer in low and moddle income countries we need to consider the possibility of unavaliable radiotheraphy service so it is important to consider other therapeutic possibilities. Patients and methods well described, Results good comparasion between the groups, toxocity, PFS and OS. Good discussion.

6. PLOS authors have the option to publish the peer review history of their article (what does this mean?). If published, this will include your full peer review and any attached files.

Reviewer #1: No

Reviewer #2: No

---

## [Author Response · Author response to Decision Letter 0]

16 Dec 2021

Dear editor,

We have received the comments on our manuscript entitled " Clinical outcome of FIGO 2018 stage IB3/IIA2 cervical cancer treated by neoadjuvant chemotherapy followed by radical surgery due to lack of radiotherapy equipment: A retrospective comparison with concurrent chemoradiotherapy " by Jing Zeng, Peisong Sun, Quanhong Ping, Shan Jiang, Yuanjing Hu. According to the comments of the reviewers, we have revised our manuscript. The revised manuscript and the detailed responses to the comments of the one reviewer are attached.

Sincerely yours,

Yuanjing Hu

Reviewer #1: For the critical analysis of the article, I suggest the evaluation of the following topics:

- Address the clinical-pathological differences between the types of cervical cancer (SCC x AC / ASC) in the discussion, as the groups present statistical differences in this regard, which may interfere with the results obtained.

Reply: Thanks for pointing out this problem. We have modified our text as advised (see Discussion, Page 17-18, line315-327) with yellow color. 

“Furthermore, 45 patients with AC/ASC had low 5-year OS. Previous studies have also suggested that the AC/ASC patients with the same FIGO stage have poorer prognostic outcomes than SCC [36, 37]. According to the present guidelines [7, 38], the therapeutic strategy recommended for SCC and AC/ASC is the same. Surgery and radiotherapy are recommended as the therapeutic schedule for early-stage cervical cancer. In our study, 38 patients were treated by CCRT, and NACT-RS only treated seven. However, a population-based analysis of about 2,773 patients with early-stage adenocarcinoma (IB-IIA) found that surgery remains the optimal local treatment modality for these patients [39]. Another retrospective study of about 3,102 patients with cervical adenocarcinoma using the Epidemiology and End Results (SEER) database also found that surgery is an independent favorable prognostic factor for early stages patients [40]. Therefore, it is significant to make new clinical treatment strategies for cervical AC/ASC.”

-Evaluate include in the discussion the recent data from the following article: DOI: 10.1038 / s41598-021-86786-y

Reply: Thanks for pointing out this problem. We have modified our text as advised (see Discussion, Page 18, line323-327) with yellow color. 

“Another retrospective study of about 3,102 patients with cervical adenocarcinoma using the Epidemiology and End Results (SEER) database also found that surgery is an independent favorable prognostic factor for early stages patients [40]. Therefore, it is significant to make new clinical treatment strategies for cervical AC/ASC.”

Reviewer #2: Well writen, considering its a common cancer in low and moddle income countries we need to consider the possibility of unavaliable radiotheraphy service so it is important to consider other therapeutic possibilities. Patients and methods well described, Results good comparasion between the groups, toxocity, PFS and OS. Good discussion.

Reply: Thanks you. This research was a small-sample retrospective study with a particular inherent bias. A larger prospective randomized trial with a multicenter is required to confirm these results.

---

## [Decision Letter · Decision Letter 1]

14 Mar 2022

Clinical  outcome of FIGO 2018 stage IB3/IIA2 cervical cancer treated by neoadjuvant chemotherapy followed by radical surgery due to lack of radiotherapy equipment: A retrospective comparison with concurrent chemoradiotherapy

PONE-D-21-13754R1

Dear Dr. Yuanjing Hu,

We’re pleased to inform you that your manuscript has been judged scientifically suitable for publication and will be formally accepted for publication once it meets all outstanding technical requirements.

Kind regards,

Ramon Andrade De Mello, MD, PhD, FACP

Academic Editor

PLOS ONE

Additional Editor Comments (optional):

Reviewers' comments:

Reviewer's Responses to Questions

**Comments to the Author**

1. If the authors have adequately addressed your comments raised in a previous round of review and you feel that this manuscript is now acceptable for publication, you may indicate that here to bypass the “Comments to the Author” section, enter your conflict of interest statement in the “Confidential to Editor” section, and submit your "Accept" recommendation.

Reviewer #1: All comments have been addressed

Reviewer #2: All comments have been addressed

2. Is the manuscript technically sound, and do the data support the conclusions?

Reviewer #1: Yes

Reviewer #2: Yes

3. Has the statistical analysis been performed appropriately and rigorously? 

Reviewer #1: Yes

Reviewer #2: Yes

4. Have the authors made all data underlying the findings in their manuscript fully available?

Reviewer #1: Yes

Reviewer #2: Yes

5. Is the manuscript presented in an intelligible fashion and written in standard English?

Reviewer #1: Yes

Reviewer #2: Yes

6. Review Comments to the Author

Reviewer #1: (No Response)

Reviewer #2: 1. In Table 2 Comparison of pathology results within NACT-RS group (line 180) could be interesting to add the pattern of response of squamous cell carcinoma and AC/ASC.

2. Line 193 missing percentage of grade 1-2 toxicities with CCRT treatment

3. Does number of partners, use of HPV vaccine has been taken into consideration in the study? Those are risk factors that could be added in Table 5 for example

4. Taking into consideration that AC/ASC are quite a different pathology of squamous, with different risk factors, prognosis, although its a small number of patients the results with these patients could be more explorated adding and extra to the article

7. PLOS authors have the option to publish the peer review history of their article (what does this mean?). If published, this will include your full peer review and any attached files.

Reviewer #1: No

Reviewer #2: No

---

## [Editor Report · Acceptance letter]

16 Mar 2022

PONE-D-21-13754R1 

Clinical outcome of FIGO 2018 stage IB3/IIA2 cervical cancer treated by neoadjuvant chemotherapy followed by radical surgery due to lack of radiotherapy equipment: A retrospective comparison with concurrent chemoradiotherapy 

Dear Dr. Hu:

I'm pleased to inform you that your manuscript has been deemed suitable for publication in PLOS ONE. Congratulations! Your manuscript is now with our production department. 

Kind regards, 

on behalf of

Dr. PLOS Manuscript Reassignment 

Staff Editor

PLOS ONE